# Metaverse Records Dataset

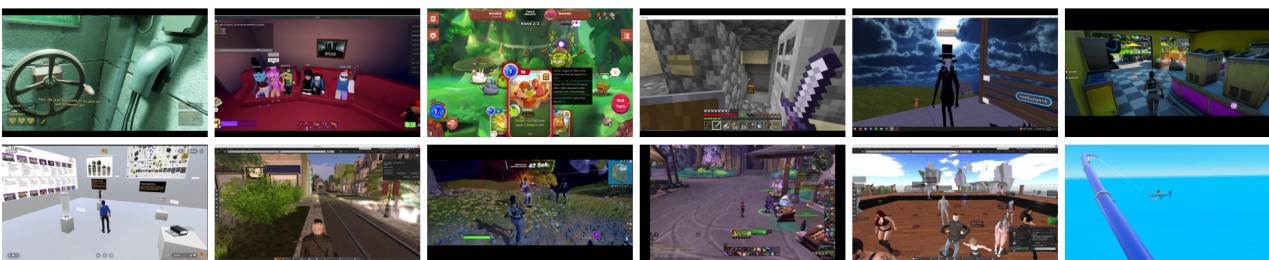

## ABSTRACT

The metaverse is an evolving field and the subject of multimedia research. In this paper, we introduce the 256-MetaverseRecords dataset, a novel and extensive collection of annotated screen recordings in the form of videos from various virtual worlds of the metaverse. We describe the process of creating the dataset, the quality criteria for the annotations, and the exploration of the dataset. We also show four experiments to evaluate the performance of different feature extraction methods for Metaverse Recordings (MVRs): MVR segmentation, audio event detection, and object and interaction detection based on this dataset. Our results demonstrate that existing methods have limitations and leave challenges in dealing with the diversity and complexity of metaverse data, and that more research is needed to develop metaverse-specific techniques. Our dataset can serve as a valuable resource for the research community and foster the development of new applications and solutions for the metaverse.

## CCS CONCEPTS

• **Computing methodologies** → **Object recognition**; **Video segmentation**; *Activity recognition and understanding*.

## KEYWORDS

Metaverse, Multimedia Retrieval, Dataset, Object Recognition

**ACM Reference Format:**
Anonymous Author(s). 2024. Metaverse Records Dataset. In *Proceedings of ACM Conference (Conference'17)*. ACM, New York, NY, USA, 7 pages. https://doi.org/XXXXXXX.XXXXXXX

## 1 INTRODUCTION

The metaverse [31] is a persistent multi-user online space. [41] describes that the metaverse is commonly based on virtual worlds, perceivable through 3D video on a screen or Virtual Reality (VR) headsets. The paper discusses current available techniques for *Metaverse Recordings (MVR)*, recordings of user sessions in the metaverse,

*Conference'17, July 2017, Washington, DC, USA*
**Unpublished working draft. Not for distribution.**
2024-04-04. ACM ISBN 978-x-xxxx-xxxx-x/YY/MM...$15.00
https://doi.org/XXXXXXX.XXXXXXX

and showed that current techniques can only partially be applied for content analysis of MVRs. Semantic understanding of MVRs requires effective content analysis techniques. To research and develop effective MVR-specific techniques, a suitable and annotated dataset for training and validation is required [3]. This paper introduces the 256-MetaverseRecords dataset for metaverse research. The dataset is publicly accessible at *anonymous*.

The remainder of the paper is structured as follows, Section 2 contains our observation results. Section 3 describes the design of the dataset, and Section 4 describes the data created. Section 5 presents the results of experiments utilizing the dataset.

## 2 STATE OF THE ART AND RELATED WORK

In this section, we present our observation results. First, we show limitation of existing datasets for *MVR*-research. Second, we describe the current metaverse market and its characteristics.

### 2.1 Overview of Datasets

A video dataset for machine learning techniques, such as object detection or human-object-detection (HOI), in virtual worlds, should fulfill the following criteria: it has to contain videos of real user session of a variety of different Metaverse virtual worlds with different look and feel and interactions. Datasets such as Div2k [4], Coco [24], or Object365 [39] contain images of the real world, which is different in visual perception.

Among the search results, we could not find a data set that matched the above criteria. The identified datasets had certain limitations: just partial things, such as heads [23, 51], objects [11], isolated from the environment [12]; interaction of driving and related sceneries only [6, 18, 22, 34]; scan data of human poses without imagery data [9]. With this research result, we decided to create a new dataset that matches our criteria.

### 2.2 Metaverse Market Overview

To select a variety of different but relevant Metaverse instances, we conducted a brief market overview and identified criteria to select virtual worlds. KZero Worldswide (former Metaversed Consulting) [28] categorizes the market into four quadrants based on

two binary criteria: Web2 vs. Web3 [48] and Browser or App access (B/A) vs. VR-based access. The first criterion is not relevant for *MVRs*, and hence, our selection. The difference of B/A or VR is relevant, because VR usually involves more devices and sensors, which theoretically gives more information to record. The perspective of the user, and hence the recording, is usually the 1st or 3rd person view, with some exceptions to Axie Infinity (2D User Interface). Furthermore, KZero grouped [28] the providers into use cases: Miscellaneous, Metaverse as a Service (MaaS), Open World, Casual Gaming, Music/Fashion/Social Hangout, User Generated Content (UGC), Education/Culture, and Real Estate.

**Table 1: Selected list of metaverse platforms from [28].**

| Metaverse | Size | Use Case | Interface |
|---|---|---|---|
| Minecraft [29] | >25m | UGC | B/A |
| Roblox [2] | >25m | Casual games | B/A |
| Fortnite [1] | >25m | Casual games | B/A |
| Zepeto [32] | 5m-25m | Social | B/A |
| Recroom [36] | 5m-25m | Social | B/A & VR |
| imvu [21] | 5m-25m | Social | B/A |
| AvakinLife [26] | 5m-25m | Social | B/A |
| Second Life [25] | 500k-1m | UGC | B/A |
| VR Chat [46] | 500k-1m | Social | VR |
| Habbo [42] | 500k-1m | Open World | B/A |
| HIBER World [19] | 500k-1m | Casual Gaming | B/A |
| Club Cooee [13] | 500k-1m | Social | B/A |
| Animal Jam [49] | 500k-1m | Casual Gaming | B/A |
| neopets Meta [33] | 500k-1m | Casual Gaming | B/A |
| Spatial [40] | 500k-1m | UGC | VR & B/A |
| Axie Infitiny [7] | 500k-1m | Casual games | B/A |
| Hytopia [20] | 500k-1m | Open World | B/A |
| Alien Worlds [14] | 500k-1m | UGC | B/A |
| Red Fox [37] | 500k-1m | Open World | VR |
| Sandbox [43] | <500k | Open World | B/A |
| Decentraland [15] | <500k | Open World | B/A |
| Horizon Worlds [27] | <500k | Social | VR |

However, there are many virtual worlds with major differences in their use- and business models. The market research company GWI has published a usage report [30], showing that Minecraft, Fortnite, The Sandbox, Horizon Worlds, Second Life, Roblox, and Decentraland are highly used platforms among makes metaverse users. KZero [28] provides information of monthly users, based on provider information. A selection is shown in Table 1. GWI and KZero differ in numbers of usage except the top 3 virtual worlds.

## 2.3 Dataset Quality Criteria

In the area of dataset annotation, achieving high data quality, consistency, and usability is important. Ensuring data quality primarily resides in the intricacies of the annotation process, an aspect extensively discussed in the literature [17, 35]. To maintain consistency, it is crucial to meticulously design and clearly articulate annotation guidelines, thus minimizing the scope for ambiguity [8, 44]. Regarding reusability, the FAIR principles — Findable, Accessible, Interoperable, and Reusable, initially conceived for data management as outlined in [50], have now been expanded to encompass dataset annotations. Together, these standards and best practices serve as the foundation for construction of a dataset that is robust, reliable, and beneficial for a wide array of academic and industrial applications.

## 3 RECORDINGS AND ANNOTATIONS

*Selection.* Given the unavailability of a suitable dataset, we undertook the task of constructing one ourselves. As metaverse recordings can differ a lot between virtual worlds, a proper variance is needed in the recordings. Therefore, we analyzed the current market and made a selection of virtual worlds for the recording.

Based on the market overview, multiple metaverses were selected, based on the criteria: at least 2 VR and 2 B/A, 2 2D, interaction and no empty worlds, cover the use cases with at least two samples, but we did not consider the KZero groups MaaS, Miscalleanous, or Real Estate world because of lack of popularity and significance to the goal of the dataset. This results in the following list: Axie-Infinity, Decentraland, Fortnite, Half Life Alyx, Meta Horizon Worlds, Minecraft, Museum Virtual Tours, Roblox, Second Life, Spatial, The Sandbox, VRchat and Word of Warcraft. Furthermore, we added two non-metaverse specific open world games for comparison reasons: World of Warcraft (B/A) and Half Life Alyx (AR).

*Use-Cases.* With this selection, various use cases were simulated: 3 UGC, 5 casual games, 2 open-world, 2 social, and 1 education/-culture; with 4 VR and 8 B/A examples. Besides 3D/VR examples, we included virtual museum tours to compare with 2D Metaverse platforms like Axie Infinity and to showcase educational use.

*Approach.* The recordings were made by screen recording in different video formats and resolutions, decided by the creators. The original instruction was to record a length of 5 minutes and should include at least 5 of the following interactions, with the virtual world, or other users. The created videos were manually annotated and the corresponding annotations were quality checked in peer review.

*Annotations.* The set of relevant annotations is defined by the following simple interaction models:

A1: walk through the virtual world, A2: bump into an obstacle (e.g. a building, an object), A3: enter a building, A4: "Waving" or some other way of "calling attention" to yourself, A5: communicate with another avatar, A6: interact with an object (e.g. play a game, watch a video), A7: to "teleport" to another place, A8: change the direction, A9: stop and walk further, A10: Chat with the system or another avatar, S0: A sound is played, S1: Message from the system, and S4: Interact with the system.

*Conversion.* The annotations were initially created in spreadsheets, later anonymized, harmonized and transferred to CSV files (interoperable and findable criteria) in a public Git repository [? ] (accessible and reusable criteria).

The following section provides an overview of the recordings and their annotations.

## 4 EXPLORATION OF THE DATASET

This section explores the details of the 256 recorded videos and more than 5963 annotations, which provide a dataset for metaverse-specific research.

The final dataset contains 256 videos of 13 virtual worlds. Table 2 shows the number of videos of each virtual world, the total length of the videos, and the minimum and maximum durations of the recordings. On average, the duration ranges from 2:05 minutes to 4:50 minutes, but there is a high variety, with an overall minimum duration of 0:23 minutes and a maximum of 22:06 minutes, and an average duration of 2:54 minutes. The resolution of the videos is between 720p and 4k. However, despite these variations, the videos provide a valuable data resource for research purposes.

**Table 2: Overview of videos in the 256-MetaverseRecords dataset**

| Metaverse | No. Videos | Total Length | Min - Max |
|---|---|---|---|
| Axie Infinity | 20 | 00:23:16 | 00:50 - 01:47 |
| Decentraland | 20 | 00:22:15 | 00:23 - 03:07 |
| Fortnite | 20 | 00:25:30 | 00:40 - 02:55 |
| Half Life Alyx | 20 | 01:08:50 | 02:16 - 04:45 |
| Horizon Worlds | 20 | 02:35:16 | 04:51 - 22:06 |
| Minecraft | 20 | 01:36:31 | 03:36 - 05:09 |
| Museum Tours | 16 | 00:23:35 | 00:35 - 01:49 |
| Roblox | 20 | 00:24:14 | 00:34 - 02:16 |
| Second Life | 20 | 00:46:14 | 01:50 - 03:02 |
| Spatial | 20 | 00:25:37 | 00:44 - 02:52 |
| The Sandbox | 20 | 01:40:18 | 05:01 - 05:02 |
| VRchat | 20 | 00:31:37 | 01:00 - 03:01 |
| Word of Warcraft | 20 | 01:39:02 | 04:41 - 05:00 |
| Averages | 19,7 | 00:57:06 | 02:05 - 04:50 |

Regarding the unique aspects of our dataset, it's important to highlight Axie Infinity and the Museum Guided Tours. Axie Infinity stands out as a 2D virtual world, offering a distinct contrast to the predominantly 3D environments in our collection. On the other hand, Museum virtual guided tours are exclusively browser-based, providing a unique approach to virtual exploration.

In terms of the exploration of annotations, the dataset contains 8110 individually checked and curated annotations. Besides the requested annotations, some editors added action types when appropriate. Table 3 provides some statistics about the initially requested action annotations.

However, there are some limitations in the annotations. One example is, that annotators entered consecutive actions in different ways, i.e. some marked a segment as walking and turning when turning somewhere inside the timeframe, while others separated these actions as three individual annotations. Another limitation is that the annotations are just time markers of the occurrence, and do not contain any bounding boxes, which limits the use of annotation, i.e. for object detection.

In general, the quality of the annotations varies through the videos but still provides a starting point for further research, or for preparing use case-specific data.

## 5 EVALUATION

In order to validate the quality of our dataset, experiments for several feature extraction methods have been performed, such as scene boundary detection, object detection, and audio detection with the dataset to assess the performance of extracting metaverse specific features or features in a metaverse.

### 5.1 Segmentation of MVRs

The MVRs generated have been deliberately limited in time, but it can be assumed that user sessions will last much longer. However, even with these shorter samples, it is noticeable that certain video segments are difficult to find. Hence, it was investigated whether the division of videos into video segments works with the available means. The experiment evaluated two methods, AWS Rekognition [5] video segmentation and ffmpeg [16], both with *MVRs* from Horizon Worlds. AWS Rekognition is purpose-build for TV video, while ffmpeg black frames detection just detects black frames, which usually occur in commercial breaks in TV programs. Movies and TV shows technically consist of camera shots, concatenated to scenes concatenated to the whole. In contrast, *MVRs* have continues view of the user, interrupted by menus and loading screens, i.e. during a teleport.

Our evaluation was conducted based on selected annotations (A3, A6, A7, S0, and S1), identified as valid segment switches. Standard metrics for video segmentation as outlined in [10] are used. However, the results presented in Table 4 indicate a low accuracy in segment recognition. This may be attributed to differences in scene boundaries of TV video and MVRs. Unlike movies and TV shows, which feature a sequence of changing shots, the Metaverse predominantly presents continuous shots. These may occasionally be interrupted by segmentations, such as teleports or menus. Specifically, in Horizon Worlds, certain teleports introduce brief black frames detectable by ffmpeg. However, the selected tools struggle to detect other transitions, leading to suboptimal segmentation outcomes.

Since actions are not similar to scenes, we described a model for scenes in *MVRs*. We defined *High-Level Metaverse Segments (HLMS)* and *Low-Level Metaverse Segments (LLMS)*. *LLMS* are divided into location stays (standing or moving in the same area), loading screens, main menus, and teleports, while *HLMS* are divided into location stays, loading screens, dialogs, and other interactions. This model is a first step and will be extended in future work. Table 5 shows the results of the metrics based on the defined segmentation model.

### 5.2 Audio event detection

In addition to visual concepts, sounds provide detectable events, maybe presenting the boundary of a scene or other events. Hence, an experiment was carried out to see if sounds can be detected. An approach is presented by Samarawickrama [38], it is also used by the Shazam algorithm [47]. An extended variant based on this approach is the comparison of audio files 'byte by byte.' By making the audio signals congruent via Dynamic Time Warping (DTW) and then comparing them based on the amplitude values. We evaluated this approach with our dataset. For that, we prepared Robolox MVRs, extracted the audio of three MVRs, and mixed sounds in the

**Table 3: Overview of the number of actions based annotations.**

| Metaverse | A1 | A2 | A3 | A4 | A5 | A6 | A7 | A8 | A9 | A10 | Sum | Avg |
|---|---|---|---|---|---|---|---|---|---|---|---|---|
| Fortnite | 149 | 39 | 38 | 10 | 6 | 81 | 10 | 53 | 12 | 0 | 398 | 39.8 |
| Roblox | 8 | 6 | 8 | 1 | 8 | 10 | 6 | 7 | 5 | 2 | 61 | 6.1 |
| Virtual Musuemguides | 0 | 0 | 0 | 0 | 0 | 0 | 0 | 0 | 0 | 0 | 0 | 0 |
| Second Life | 251 | 44 | 28 | 10 | 7 | 240 | 22 | 152 | 186 | 9 | 949 | 94.9 |
| Axie Infinity | 0 | 0 | 0 | 0 | 0 | 0 | 0 | 0 | 0 | 0 | 0 | 0 |
| Decentraland | 24 | 23 | 24 | 32 | 5 | 29 | 12 | 113 | 115 | 0 | 377 | 37.7 |
| Minecraft | 127 | 12 | 17 | 6 | 0 | 96 | 6 | 164 | 20 | 10 | 458 | 45.8 |
| The Sandbox | 42 | 21 | 17 | 10 | 0 | 28 | 8 | 22 | 21 | 21 | 190 | 19 |
| Spatial.io | 203 | 38 | 24 | 36 | 30 | 62 | 34 | 60 | 37 | 20 | 544 | 54.4 |
| Meta Horizon Worlds | 230 | 22 | 15 | 8 | 2 | 224 | 104 | 51 | 22 | 0 | 678 | 67.8 |
| VRchat | 53 | 11 | 9 | 17 | 27 | 55 | 17 | 30 | 28 | 8 | 255 | 25.5 |
| Half Life Alyx | 163 | 24 | 20 | 50 | 31 | 684 | 246 | 369 | 0 | 63 | 1650 | 165 |
| World of Warcraft | 96 | 11 | 7 | 11 | 8 | 70 | 7 | 151 | 30 | 12 | 403 | 40.3 |
| Sum | 1346 | 251 | 207 | 191 | 124 | 1579 | 472 | 1172 | 476 | 145 | 5963 | |
| Min | 0.0 | 0.0 | 0.0 | 0.0 | 0.0 | 0.0 | 0.0 | 0.0 | 0.0 | 0.0 | 0.0 | |
| Max | 251.0 | 44.0 | 38.0 | 50.0 | 31.0 | 684.0 | 246.0 | 369.0 | 186.0 | 63.0 | 1650.0 | |
| Avg | 103.5 | 19.3 | 15.9 | 14.7 | 9.5 | 121.5 | 36.3 | 90.2 | 36.6 | 11.2 | 458.7 | |
| Std | 89.9 | 14.5 | 11.1 | 15.3 | 11.7 | 185.5 | 68.6 | 102.7 | 54.0 | 17.3 | 449.2 | |

**Table 4: Evaluation AWS Rekognition and ffmpeg on action annotations.**

| Method | AP | Recall (trs 0.5) | Recall@3s (trs 0.5) |
|---|---|---|---|
| AWS Rekognition | 0.0126 | 0.0497 | 0.3920 |
| ffmpeg black frame | 0.0186 | 0.0259 | 0.0884 |

**Table 5: Evaluation AWS Rekognition and ffmpeg on *HLMS* and *LLMS*.**

| Method | AP | Recall (trs 0.5) | Recall@3s (trs 0.5) |
|---|---|---|---|
| AWS Rekognition (Confidence 0.6) | 0.089 | 0.387 | 0.536 |
| ffmpeg black frame (filter 0.1) | 0.1 | 0.595 | 0.697 |

audio. As parameters, the loudness threshold (lt) was set slightly below the calculated Sum per Size (SpS), the average loudness of the analyzed audio file $lt = SpS - 0.2$, a fixed resolution value ($resolution = 100$). This method achieved an accuracy of 64.86%, precision of 96.00%, recall 66.67%, and a $F_1$ score of 0.7869. Further manual fine-tuning achieved better values, shown in Table 6, with just four false negative (FN) occurrences, marked with x in the table, and one false positive (FP), marked as FP.

In conclusion, DTW audio event recognition can work well with a suitable audio sample. However, the results from DTW are not sufficient for an effective functionality of such recognition techniques.

## 5.3 Classification and Object Detection with Google Vertex

This experiment researched the classification of scenes and detection of objects in Axie Infinity. In this experiment, the Google Vertex API with custom model training was used. Based on the 2D game Axie Infinity videos, a training set of 370 images was created to detect labels. These images were manually assigned to the custom labels 'fight_adventureMode', 'fight_adventureMode_attack', 'fight_adventureMode_enemiesTurn', 'fight_arenaMode', 'fight_arenaMode_attack', 'fight_arenaMode_enemiesTurn',

'navigation_adventureMode' and 'navigation_menu'. The labels correspond to simple interactions that occur while playing Axie Infinity. For each label, 50 screenshots have been selected, except the label 'navigation_adventureMode' has only been assigned to 20 screenshots.

For the experiment, a sample of 300 images was considered for label recognition. 150 images are subject to the analysis for object recognition using Google Vertex AI.

In the total images there are 40 images of the class 'fight_adventureMode', 'fight_adventureMode_attack', 'fight_adventureMode_enemiesTurn', 'fight_arenaMode', 'fight_arenaMode_attack', 'fight_arenaMode_enemiesTurn', and 'navigation_menu'. In addition, 20 images of the class 'navigation_adventureMode' are included. Some images were automatically removed by Vertex AI during the training process.

Table 7 shows the recall of the detected label classes. In general, a recall of 77.33% and a precision of 81.69% were achieved with custom training.

In addition to the labeling of the images, a custom object detection was trained and evaluated. The corresponding objects have been assigned with the labels 'axie_own', 'axie_enemy' and 'creature_enemy'. Of the 150 images, 100% were of interest for detecting the objects with the label 'axie_own'. 50% of the images are each tested for objects with the labels 'axie enemy' and 'creature_enemy'.

The recall for the classes is axie_own 82.67%, axie_enemy 46.67%, and creature_enemy 50.67%.

For the detection of interactions and labels, the experiment produced an overall correct recall of 77.33%. However, specifically for the labels in 'fight_arenaMode' there was a lower recall of 61.16%, while the labels for other areas showed very reliable results, especially 100% for 'navigation'. Regarding 'fight_adventureMode', labels were correctly assigned, but not always to the specific expected label, possibly due to a non-expressive model. A peculiarity was found in 'arenaMode', where the recall was poor, and differences in image representations could be the main reason. It seems that Google's AI in VertexAI mainly works with the dominant features, such as the background, which could explain some wrong decisions. Overall, however, the approach is feasible.

**Table 6: Optimal values for recognition of a sound in audio files as time with the values of lowest warp distance (lwd). The sound is detected with the settings loudness threshold (lt), and distance threshold (dt).**

| Audio File | SpS | Time (s) | lwd | dt | lt | Time (s) | lwd | dt | lt | Time (s) | lwd | dt | lt | Time (s) | lwd | dt | lt |
|---|---|---|---|---|---|---|---|---|---|---|---|---|---|---|---|---|---|
| - | | | Sound: ArpShooter1 | | | | Sound: BottleCrinkle | | | | Sound: ChipOpaSid5 | | | | Sound: ChipOpaSid6 | | |
| Adopt Me | 2.06 | 3.3 | 187147 | 190000 | 12 | 6.6 | x | | | 17.3 | 47106 | 80000 | 5 | 0.37 | 106643 | 110000 | 10 |
| | | 11.6 | 175250 | | | 34.3 | x | | | 46.8 | 73465 | | | 10.36 | 77315 | | |
| | | 34.8 | 141855 | | | 50 | x | | | 58.2 | 68943 | | | 23.5 | 77315 | | |
| Pizzastore | 2.4 | 3.3 | 237460 | 240000 | 11 | 6.6 | 57705 | 75000 | 2.4 | 17.3 | 62251 | 70000 | 5 | 0.37 | 74490 | 85000 | 10 |
| | | 11.6 | 86460 | | | 34.3 | 66731 | | | 46.8 | 60023 | | | 10.36 | 82241 | | |
| | | 34.8 | 119178 | | | 50 | x | | | 58.2 | 67094 | | | 23.5 | 69817 | | |
| Prisonbreak | 0.75 | 3.3 | 196568 | 200000 | 11 | 6.6 | 62998 | 70000 | 2.4 | 17.3 | 66220 | 70000 | 5 | 0.37 | 69270 | 75000 | 10 |
| | | | | | | 24 | FP | | | 46.8 | 56157 | | | 10.36 | 74145 | | |
| | | 11.6 | 116257 | | | 34.3 | 28381 | | | 58.2 | 61526 | | | 23.5 | 68607 | | |
| | | 34.8 | 106156 | | | 50 | 471515 | | | | | | | | | | |

**Table 7: Evaluation results for Google Vertex detection of Axie Infinity objects.**

| Label | Recall |
|---|---|
| fight_adventureMode | 0.5526 |
| fight_adventureMode_attack | 0.9524 |
| fight_adventureMode_enemiesTurn | 0.9500 |
| fight_arenaMode | 0.6000 |
| fight_arenaMode_attack | 0.7000 |
| fight_arenaMode_enemiesTurn | 0.5366 |
| navigation_adventureMode | 1.0000 |
| navigation_menu | 1.0000 |

The relatively good results of interaction recognition are contrasted by the poor results of the object recognition. An overall recall of 65.67% is far below the expected result of 75%. In general, the detection of enemy creatures seems to be a major problem for object detection. A recall of 46.67% for objects with the label 'axie_enemy' and a recall of 50.67% for the label 'creature_enemy' is a big difference to the quite good result of 82.67% for objects from the label 'axie_own'. While processing the results, several problems were noticed, which could explain the relatively poor results.

## 5.4 Avatar and Interaction Detection of MVRs

A final experiment was conducted to detect interaction with objects in a VR virtual world VR-Chat. The selected interaction is 'eating chips'. Two different methods were evaluated, both employing YOLOv8 [45] nano, pre-trained with Coco-128 [24] and then prepared with transfer learning of annotated images. The prepared training data contain images with 59 instances of the class "chips bowls", 33 instances of the class "Avatar", and 31 instances of the class "eating_animation", which is played if chips have been eaten. The method tried to identify the combination of the eating_animation and chipsbowl, which counts as the targeted interaction. The second method is to detection of the interaction "eating chips" based on the distance between the objects avatars and chips bowl.

For the evaluation of the first method, a test data set of 18 short videos was created. This test dataset contains 7 negative test videos, i.e. videos in which there is no interaction with the chip dish, in order to be able to check whether the interaction detection is falsely triggered. In 10 of these 18 test videos, semi-human avatars can be seen, as they are often selected by users in VR chat (fairies, human avatars with animal ears and/or animal tails), 5 of the videos contain purely human avatars, and 3 videos contain non-human avatars.

The test set contained a total of 52 disappearing chip animations in combination with a visible chips bowl. With a $confidence\_threshold = 0.5$, the custom-trained YOLOv8 correctly recognized 19, true positive (TP), interaction, the remaining 33 interactions were not recognized, hence, FN, and 11 were FP. This results with an accuracy of 37.01%, a precision of 63.33%, and a recall of 36.54%. This results in an $F_1$ score of 0.4634. Possible reasons for FP are incorrect detection of objects with similar colors as the chips bowls, i.e. usernames.

For the second method, based on the distance between the objects, the annotations could not be used for a verification. Hence, a manual check of all detections was made. Table 8 shows the evaluation results. The recognition of the object "avatar" was taken into account when assessing the results, thus the results were grouped by divided into three categories: 'human', 'semi-human' and 'non-human' avatars in the interaction.

**Table 8: Evaluation results based on distance.**

| Class | TP | FP | FN | TN |
|---|---|---|---|---|
| human_avatars | 253 | 62 | 382 | 750 |
| | | 100% POV | 68% POV | |
| | | | 32% n.d. | 64% n.d. |
| non-human_avatars | 108 | 80 | 221 | 695 |
| | | 100% POV | 25% POV | |
| | | | 75% n.d. | 98% n.d. |
| semi-human_avatars | 1133 | 226 | 561 | 1351 |
| | | 100% POV | 48% POV | |
| | | | 52% n.d. | 25% n.d. |
| total | 1494 | 368 | 1164 | 2796 |
| | | 100% POV | 47% POV | |
| | | | 53% n.d. | 54% n.d. |

Furthermore, the results are separated by Point-Of-View (POV) and not detected (n.d.), denoting a perspective error in the detection, for example, if a person is standing close to the chips bowl but is not recognized as being close by the checkDistance() function. N.d. refers to no detection, which means that the avatar was not recognized. The table shows that FP detections occur due to perspective

errors. Although the TP detection returns a "correct" result, 54% of this result is due to the fact that the person was not detected and therefore no other result could have been output. The FN results are caused by both the perspective (47%) and the non-detection of the person (53%).

The human avatars have a $precision_{human\_avatars}$ = 80.32%, a $recall_{human\_avatars}$ = 39.84% and an $F1_{human\_avatars}$ = 53.26. This means that if the program reports an interaction with human avatars based on distance, there is a probability of about 80% that it is an interaction, but only about 39.8% of the interactions are recognized at all. For the available data on non-human avatars, this results in a $precision_{non-human\_avatars}$ = 57.45%, a $recall_{non-human\_avatars}$ = 32.83% and consequently an $F1_{non-human\_avatars}$ = 41.78%. For the semi-human avatars, this results in a $precision_{semi-human\_avatars}$ = 83.37%, a $sensitivity_{semi-human\_avatars}$ = 66.88% and an $F1_{semi-human\_avatars}$ = 74.22%. Surprisingly, the data set with the semi-human avatars thus shows the best performance in all areas, compared to human avatars and also compared to completely non-human avatars. Therefore, the current implementation achieves an $precision_{total}$ = 80.2% and an overall $recall_{total}$ = 56.2%, an $F1_{total}$ = 66.03%.

The two main causes of the recognition problems are very clear in this case. First, there are perspective errors due to the checkDistance() method, which in this case only checks whether the bounding boxes of the objects overlap using the upper left coordinate and the lower right coordinate of the bounding boxes. This leads to interactions being falsely reported if a person walks behind the chips bowl in the background and the boxes therefore overlap, or, to non-detection of interactions if the person is directly next to the chip bowl, but the bounding boxes are next to each other instead of overlapping.

The second main source of error is the recognition of avatars as a person. Depending on the avatar, it seems to be difficult for YOLOv8 to recognize this person as such, as some avatars have non-human attributes, and some are no longer human at all. Due to the diversity of selectable avatars in VR chat, which can be animals, cuddly cushions or abstract images (it is possible to walk around as a 2-dimensional image), this represents an open challenge,

Contrary to the expectation that detection based on the combination should work better, as it is not dependent on whether semi- to non-human avatars are recognized, interaction detection based on distance in the current implementation has significantly higher precision and recall than interaction detection based on the combination. YOLOv8 showed potential for avatar detection, which can be optimized in further work, in particular for non-human avatars and further metaverses than VR-Chat.

## 5.5 Discussion

We showed three exploratory experiments, which demonstrate that the novel dataset provides a useful data source for machine learning experiments specific to the metaverse. We further outlined that the segmentation of videos requires further metaverse-specific methods. The detection of audio events with the selected methods makes it clear that more research is needed. The detection of objects and avatars in the videos shows promising results but requires further research to refine the methods and additional or higher data quality based on the videos. Furthermore, the experiments and their results show the importance of a valid and appropriate dataset for research in the area of metaverse.

## 6 CONCLUSION

The presented dataset provides 256 metaverse recordings of a balanced mix of virtual worlds. The recordings range from 00:50 to 22:06 minutes, with a total length of 12.37 hours. The dataset also provides 5963 annotations for interactions in the videos. The annotations are limited to time markers, no bounding boxes or further details are provided. Hence, the videos can be used for machine learning experiments, but more high-quality annotations are required. However, the data set provides a base for further work in the field of metaverse-specific content analysis techniques.

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

Received

