# OpenReview forum: "256 Metaverse Recording Dataset"
_acmmm.org/ACMMM/2024/Conference — MM2024 Poster_

### Official Review · Reviewer_nRym · 2024-05-22

**Rating:** 4
**Confidence:** 2

**Summary:**

It is a fairly well written paper in which the authors present a novel dataset to provides the XR research community with a useful data source for machine learning-based experiments specific to the metaverse.

**Strengths:**

The paper is quite well written, reasonably structured, and it is easy to follow.
The presented novel dataset contains 12.37 hours in 256 recordings of some selected virtual worlds, with a high number of annotations for the interactions in the recordings. Four experiments have been conducted to evaluate the performance of some feature extraction methods for the recordings, with interesting results regarding their limitations. Although it needs to be improved, it will be a valuable resource for the research community.

**Limitations:**

The description of the conducted experiments/tests to evaluate the use of the dataset is too brief. More details should be provided and the obtained results should be discussed a bit more.

The conducted experiments/tests (e.g., the ones to obtain the results shown in tables 4 to 7) should be explained a bit more and the
 obtained results should also be discussed in more detail in the paper.

Please, although it can be deduced, explain in the text what ‘k’ and ‘m’ mean in table 1 , ‘AP’ in tables 4 and 5, and TN in Table 8.

Some typos or possible mistakes to be corrected:
- Revise the first sentence in the abstract. It is wrong
- (many times) data set -> dataset
- Revise this sentence: highly used platforms among ¿makes? metaverse users.
- Miscalleanous -> Miscellaneous
- Half Life Alyx (AR) -> Half Life Alyx (VR)
- A Word is missing in this senence: “This results in the following list:”. Maybe “This results ARE in the following list”
- This section explores the details of the 256 recorded videos and more than 5963 annotations, -> ... more than 8100 annotations,
- For each label, 50 screenshots have been selected, except the label ’navigation_adventureMode’ has only been assigned to 20 screenshots. -> ... except the label ’navigation_adventureMode’ THAT has only been assigned to 20 screenshots.
- Revise this sentence: The second method is to detection of the interaction "eating chips"
- The test set -> The test dataset
- the results were ¿grouped by divided into? three categories
- achieves an 𝑝𝑟𝑒𝑐𝑖𝑠𝑖𝑜𝑛𝑡𝑜𝑡𝑎𝑙 -> achieves A 𝑝𝑟𝑒𝑐𝑖𝑠𝑖𝑜𝑛𝑡𝑜𝑡𝑎𝑙

**Suitability:**

2

---

### Official Review · Reviewer_MinJ · 2024-05-25

**Rating:** 4
**Confidence:** 3

**Summary:**

[Objective] The work aims to address the challenges in semantic understanding of Metaverse Recordings (MVRs) by introducing the 256-MetaverseRecords dataset. It is a collection of annotated screen recordings from various virtual worlds. [Findings] The authors conducted experiments to evaluate different feature extraction methods for MVRs, including segmentation, audio event detection, and object and interaction detection.  [Contribution] The contribution lies in the creation of a valuable dataset that can foster research in the metaverse domain, although further comparison with related work would enhance the demonstration of the work’s uniqueness.

**Strengths:**

[Strengths]
- The work is well-written and demonstrates an easy flow
- Comprehensive dataset creation with a focus on diversity and complexity of metaverse data.

**Limitations:**

[Points of Improvement]
- Clarification on the annotation process and consistency across different virtual worlds.
- Lacks of better comparison with related work
- Expansion on the segmentation model to include more granular details.
- Improvement in object detection accuracy, particularly for non-human avatars.




[Typos suggestions using HTML tags <ins> and <del>]
- <del>ffmpeg</del><ins>FFmpeg</ins>
- results. First, we show <ins>the</ins> limitation<ins>s</ins> of existing datasets for MVR<ins></ins><del>-</del>
- overview and identified criteria <del>to</del><ins>for</ins>select virtual worlds.
- GWI and KZero differ in numbers of usage except<ins>for</ins>
- as the foundation for <ins>the</ins>
- we analyzed the current market and <del>made a selection of</del><ins>select</ins>
- multiple metaverses were selected<del>,</del> based on the criteria
- KZero groups MaaS, Misc<ins>e</ins><del>a</del>lleanous,
- The created videos were manually annotated<ins>,</ins>
- AWS Rekognition is purpose-buil<ins>t</ins><del>d</del> for TV video, while
- VR virtual world VR<del>-C</del><ins>c</ins>hat. The selected
- FN, and 11 were FP. Th<ins>ese</ins><del>is</del> results <del>with</del><ins>have</ins> an accuracy of 37.01%

**Suitability:**

3

---

### Official Review · Reviewer_ZfWi · 2024-05-26

**Rating:** 5
**Confidence:** 3

**Summary:**

The authors collected 256 video sequences from the metaverse and built the metaverse records dataset.

The metaverse records dataset contains video sequences from multiple metaverse platforms such as Minecraft, Roblox, VR Chat, etc.

Each record in the dataset is annotated.

**Strengths:**

- It is the first dataset that contains systematically collected video sequences from metaverse
- The authors demonstrates that some potential open problems on the dataset

**Limitations:**

- Maybe the video sequences can also be recorded in 360 degree format

**Suitability:**

3

---

### Official Review · Reviewer_H6u2 · 2024-05-26

**Rating:** 2
**Confidence:** 3

**Summary:**

This paper introduces the 256-MetaverseRecords dataset, a collection of annotated screen recordings from various virtual worlds. The creation process, annotation quality, and dataset exploration are introduced in the paper. Four experiments evaluate feature extraction methods for segmentation, audio event detection, and object and interaction detection. Results show current methods have limitations, highlighting the need for metaverse-specific techniques, and this dataset aims to support further research and development.

**Strengths:**

1. The dataset includes data from various virtual worlds with detailed interaction annotations, offering significant opportunities for applying machine learning to metaverse data. These interaction annotations capture a wide range of user actions and behaviors within virtual environments. This granularity allows machine learning models to learn and understand fundamental actions within metaverse data, enabling the development of more sophisticated and context-aware algorithms.
2. By providing a set of annotated interactions, the dataset supports tasks such as user behavior prediction. Moreover, the comprehensive nature of these annotations can help in training models to recognize and respond to complex interaction patterns, ultimately contributing to the advancement of technologies and applications in the metaverse.

**Limitations:**

1. Dataset Size: The dataset is relatively small, consisting of only 256 recordings with a total duration of 12.37 hours. It is necessary to demonstrate that this dataset is sufficient for training robust machine learning models, as larger datasets typically lead to better generalization and performance.
2. Annotation Process and Quality: The paper lacks detailed information about the annotation process and the qualifications of the annotators. Understanding the methodology behind the annotations and the expertise of those who performed them is crucial for assessing the reliability and accuracy of the dataset.
3. Annotation Detail: The annotations are limited to time markers, without any bounding boxes or further contextual details. This limits the dataset's usefulness for more detailed analysis and machine learning experiments that require precise localization and context of interactions.

**Suitability:**

3

---

### Meta-Review · Area_Chair_nbwL · 2024-07-03

**Recommendation:** Accept (Poster)
**Confidence:** 5

**Metareview:**

The reviewers conclude on the merit of this work. Given the feedback, this would fit a poster presentation.